# Whole-Blood Gene Expression Profiles Associated with Mortality in Community-Acquired Pneumonia

**DOI:** 10.3390/biomedicines11020429

**Published:** 2023-02-01

**Authors:** Diego Viasus, Antonella F. Simonetti, Lara Nonell, Oscar Vidal, Yolanda Meije, Lucía Ortega, Magdalena Arnal, Marta Bódalo-Torruella, Montserrat Sierra, Alexander Rombauts, Gabriela Abelenda-Alonso, Gemma Blanchart, Carlota Gudiol, Jordi Carratalà

**Affiliations:** 1Department of Medicine, Division of Health Sciences, Universidad del Norte and Hospital Universidad del Norte, Barranquilla 081001, Colombia; 2Department of Internal Medicine, Consorci Sanitari Alt Penedès-Garraf, 08720 Sant Pere de Ribes, Spain; 3Centro de Investigación Biomédica en Red de Enfermedades Infecciosas (CIBERINFEC), Institulo de Salud Carlos III, 28029 Madrid, Spain; 4MARGenomics, Hospital del Mar Medical Research Institute (IMIM), 08003 Barcelona, Spain; 5Unit of Infectious Disease, Department of Internal Medicine, Hospital de Barcelona—Societat Cooperativa d’Instal·lacions Assistencials Sanitàries (SCIAS), 08029 Barcelona, Spain; 6Microbiology Unit, Department of Clinical Laboratory, Hospital de Barcelona—Societat Cooperativa d’Instal·lacions Assistencials Sanitàries (SCIAS), 08029 Barcelona, Spain; 7Department of Infectious Diseases, Bellvitge University Hospital—Bellvitge Biomedical Research Institute (IDIBELL), 08907 Barcelona, Spain; 8Cardiovascular Risk and Nutrition Research Group, Hospital del Mar Medical Research Institute (IMIM), 08003 Barcelona, Spain; 9Department of Clinical Sciences, University of Barcelona, 08907 Barcelona, Spain

**Keywords:** community-acquired pneumonia, mortality, gene expression profile, gene set enrichment analysis

## Abstract

(1) Background: Information regarding gene expression profiles and the prognosis of community-acquired pneumonia (CAP) is scarce. We aimed to examine the differences in the gene expression profiles in peripheral blood at hospital admission between patients with CAP who died during hospitalization and those who survived. (2) Methods: This is a multicenter study of nonimmunosuppressed adult patients who required hospitalization for CAP. Whole blood samples were obtained within 24 h of admission for genome-expression-profile analysis. Gene expression profiling identified both differentially expressed genes and enriched gene sets. (3) Results: A total of 198 samples from adult patients who required hospitalization for CAP were processed, of which 13 were from patients who died. Comparison of gene expression between patients who died and those who survived yielded 49 differentially expressed genes, 36 of which were upregulated and 13 downregulated. Gene set enrichment analysis (GSEA) identified four positively enriched gene sets in survivors, mainly associated with the interferon-alpha response, apoptosis, and sex hormone pathways. Similarly, GSEA identified seven positively enriched gene sets, associated with the oxidative stress, endoplasmic reticulum stress, oxidative phosphorylation, and angiogenesis pathways, in the patients who died. Protein–protein-interaction-network analysis identified *FOS*, *CDC42*, *SLC26A10*, *EIF4G2*, *CCND3*, *ASXL1*, *UBE2S*, and *AURKA* as the main gene hubs. (4) Conclusions: We found differences in gene expression profiles at hospital admission between CAP patients who died and those who survived. Our findings may help to identify novel candidate pathways and targets for potential intervention and biomarkers for risk stratification.

## 1. Introduction

Community-acquired pneumonia (CAP) is a public health problem worldwide and continues to be associated with high health costs, morbidity, and mortality. Over the coming years, the overall burden of CAP is likely to rise as its incidence and the number of elderly people increase [1]. Recent studies have found overall mortality rates of 5% to 15% among hospitalized patients with CAP, and mortality in the subset of patients who require intensive care unit (ICU) admission may be as high as 30%: a rate that matches those of other known medical-emergency diseases, such as ST-elevation myocardial infarction [2]. Most cases of CAP occur when its organisms translocate from the nasopharynx to the lungs. Infection takes place when there has been exposure to a large inoculum or a virulent microorganism and/or the host defenses are impaired.

Studies have stressed the importance of host features in the prognosis of CAP, including inflammatory response, susceptibility to specific pathogens, genome, and metabolic condition [3]. In this regard, the information derived from gene expression studies may broaden our understanding of the complexity of the immune response via identification of novel candidate pathways and targets for potential intervention, discovery of novel candidate diagnostic and stratification biomarkers, and our increased ability to categorize patients into clinically relevant expression-based subclasses. At present, however, information regarding the gene expression profile in CAP is scarce; most published studies have been performed in animal models or focused on patients with sepsis in ICU or patients with specific etiologies, such as pneumococcal pneumonia [4,5,6]. 

In the present study, we comprehensively examined differences in gene expression profiles in peripheral blood at hospital admission between CAP patients who subsequently died and those who survived in order to identify genes and related pathways that not only provide information about pathophysiology but may also serve as prognostic biomarkers.

## 2. Materials and Methods

This study was conducted at two university hospitals for adults in Barcelona, Spain. Nonimmunosuppressed adult patients who required hospital admission for CAP from May 2015 through January 2017 were prospectively recruited and followed up on. All patients included in this study were enrolled within 24 h of hospital admission. This study was approved by the Ethics Committee of the Bellvitge University Hospital (approval code: PR158/14; 6 November 2014). The relevant data and protocols comply with the minimum information about a microarray experiment (MIAME) guidelines [7].

Patients were classified as having CAP if they had an infiltrate on a chest radiograph plus acute illness associated with two or more of the following signs and symptoms: a new cough with or without sputum production, pleuritic chest pain, dyspnea, fever or hypothermia, altered breath sounds on auscultation, or leukocytosis or leukopenia. Patients were seen by the clinical investigators, who recorded data in a computer-assisted protocol, daily during their hospital stays. Demographic characteristics, comorbidities, causative organisms, antibiotic susceptibilities, biochemical analysis, empirical antibiotic therapy, and outcomes were recorded. Patients with neutropenia, solid organ transplantation, antineoplastic chemotherapy, acquired immunodeficiency syndrome (AIDS), current corticosteroid therapy (≥20 mg prednisone/d or equivalent), and pregnancy at admission were excluded. 

### 2.1. RNA Extraction

Whole blood samples (2.5 mL) were obtained within 24 h of hospital admission for genome expression profile analysis. Total RNA was isolated from these whole blood samples using the PaxGene™ blood RNA system (PreAnalytiX, Qiagen/Becton Dickson, Hombrechtikon, Switzerland) in accordance with the manufacturer’s specifications. Extracted RNA was stored at minus 80 degrees Celsius until expression profiling. RNA quantification was performed using a spectrophotometer (NanoDrop Technologies, Wilmington, DE, USA) and RNA quality was assessed using an Agilent Bioanalyzer 2100 slide. 

### 2.2. Gene Expression Profiles via Microarrays

Gene expression microarrays were performed at the MARGenomics facilities of the Hospital del Mar Medical Research Institute (IMIM). RNA samples were amplified and labeled with a GeneChip WT PLUS Reagent kit and hybridized to a Clariom S Human array (Affymetrix, Santa Clara, CA, USA) in a GeneChip Hybridization Oven 640. The washing and scanning steps were performed using the Expression Wash, Stain and Scan Kit and the GeneChip System of Affymetrix (GeneChip Fluidics Station 450 and GeneChip Scanner 3000 7G). After quality control, the raw data were background-corrected, quantile-normalized, and summarized to a gene level using the robust multichip average (RMA) [8], obtaining a total of 20,893 transcripts, excluding controls. NetAffx 36 annotations that corresponded to the hg38 human genome version were used to summarize data into transcript clusters and annotate all of the transcripts analyzed. The microarray data from the present project were deposited in the Gene Expression Omnibus of the National Center for Biotechnology Information (NCBI) under accession number GSE188309.

### 2.3. Functional Analysis of Expression Data

Functional analysis was performed with gene set enrichment analysis (GSEA) [9,10]. To obtain a summary of the biological states and processes underlying our analysis, we used the Hallmark gene-set collection defined by the Molecular Signatures Database (mSigDB), UC San Diego and Broad Institute, USA (http://www.gsea-msigdb.org/gsea/msigdb). In addition, NetworkAnalyst 3.0, Canada, (https://www.networkanalyst.ca/) and IntAct Molecular Interaction Database, European Molecular Biology Laboratory, EMBL’s European Bioinformatics Institute, UK, (https://wwwdev.ebi.ac.uk/intact) were used to construct the protein–protein interaction (PPI) network. These tools allow the generation of an enrichment network in which nodes that may be relevant in the analysis of gene expression can be visualized.

### 2.4. Statistical Analysis

To obtain the list of differentially expressed genes in the various analyses, a double strategy was followed. As a first approach, linear models for microarray (limma analysis) were used to detect differentially expressed genes between the conditions, including a variable batch to adjust for batch differences [11]. Next, using the same linear models, a subsampling strategy was applied. In brief, 1000 models were generated for each analysis, using a random and balanced subsample of the cases (representing between 55% and 77% of cases, depending on the model). To adjust the possible batch influence, a variable batch was also included in this model. The genes most frequently selected as being differentially expressed were defined as top differentially expressed. Genes with a *p*-value of less than 0.05 were selected as significant. GSEA results were considered statistically significant when a gene set had a *p*-value of less than 0.05 and the false discovery rate (FDR) was less than 0.25, following the Broad Institute FAQ guidance. Moreover, expression data for each gene and prognosis were recorded for each patient, and survival curves were generated using survminer R packages. Samples above the median expression were considered highly expressed whereas samples below the median were considered low-expressed. All data analyses were performed in R (version 3.4.2).

## 3. Results

### 3.1. Characteristics of the Cohort

During the study period, 228 consecutive nonimmunosuppressed CAP patients were admitted to the hospital, of whom 18 (7.9%) died during hospitalization. The main sociodemographic and clinical features and laboratory findings thereof are shown in Table 1. Most patients were older than 65 years (69.7%), and 154 (67.5%) presented comorbidities, mainly chronic pulmonary and cardiac diseases and diabetes mellitus. Nearly half of the patients had respiratory failure at admission. Regarding etiology, *Streptococcus pneumoniae* was the most frequent causative pathogen. Most patients (64%) were classified as high-risk (pneumonia severity index (PSI) IV–V). The causes of mortality were respiratory failure, multiorgan dysfunction, and septic shock.

### 3.2. Differentially Expressed Genes between CAP Patients Who Died and Those Who Survived

A total of 198 samples were processed, 13 of them from patients who died. Missing samples were not processed due to quality or sample-quantity issues. Table 2 shows the list of genes that were differentially expressed in CAP patients who died in relation to in those who survived based on the lowest *p*-value. Analysis yielded 49 differentially expressed genes, 36 of which were upregulated and 13 downregulated in circulation of whole-blood cells. Figure 1 shows the heat map of the differentially expressed genes. In addition, the analyses of the expression data for each gene and prognosis indicated that low expression of the genes *SPRYD3*, *ESPLI*, and *HHIPL2* and high expression of the gene *PLXNA1* were significantly related to survival in CAP patients (Figure 2). Other genes whose expression tended to be related to survival were *COQ6*, *METTL20*, *PGAP2*, *DIXDC1*, *NRL*, and *ADCK4*. The curves of these analyses are shown in Appendix A.

### 3.3. Functional Analysis

GSEA was used to identify differentially expressed gene sets. A total of four out of fifty gene sets were positively enriched in the patients who died, and seven out of fifty gene sets were positively enriched in the patients who survived (NOM *p*-value < 0.05 and/or FDR < 25%) (Table 3 and Figure 3). Enrichment plots of the gene sets are shown in Appendix A. Moreover, PPI analysis with NetworkAnalyst 3.0 and IntAct found FOS, CDC42, SLC26A10, EIF4G2, CCND3, ASXL1, UBE2S, and AURKA to be the main gene hubs (Figure 4). The names, abbreviations, and functions of these gene hubs are shown in Table 4.

## 4. Discussion

In this study, we identified the whole-blood gene expression profile associated with mortality in CAP patients. Functional enrichment analysis showed differences in gene expression profiles at hospital admission between CAP patients who died and those who survived, mainly regarding interferon alpha response, oxidative stress, apoptosis, endoplasmic reticulum stress, sex hormones, and angiogenesis pathways.

Studies that have evaluated gene expression profiles associated with severity or mortality in CAP are scarce. Hopp et al. [4] reported that in blood transcriptomes from septic patients in the ICU, CAP severity was associated mainly with immune dysregulation (T cell immune suppression, chemokine receptor deactivation, and macrophage polarization). Similarly, having evaluated patterns of gene expression in blood mononuclear cells from patients with sepsis secondary to CAP, Severino et al. [5] found that differences in oxidative phosphorylation seemed to be associated with prognosis at the time of patient enrollment. In addition, after comparing samples at admission and during follow-up, those authors found that gene expression profiles differed between survivors and nonsurvivors, with decreased expression of genes related to immune functions. Our study differs from these previously published papers with regard to objectives, inclusion criteria, and methods for assessing gene expression (mononuclear cells vs. whole blood). In fact, the differences between studies may provide insights into the distinct characteristics of the host response during CAP. 

Furthermore, studies have also evaluated the performance of gene expression profiling in predicting prognosis in heterogeneous cohorts of sepsis patients. Hu et al. [12] performed a bioinformatic analysis of gene expression profiles for prognosis in patients with septic shock. Those researchers found that differentially expressed genes between septic shock patients and controls were primarily involved in the MAPK, tumor necrosis factor, HIF-1, and insulin signaling pathways. Six genes were identified to be positively correlated with prognosis in patients with septic shock. Another study found that sepsis response transcriptomic signatures (SRSs) can define subgroups of patients related to a sepsis outcome [13]. Cell death, apoptosis, necrosis, T cell activation, and endotoxin tolerance are enriched biological functions that pertain to SRSs in intra-abdominal and respiratory infection. SRSs is associated with higher early mortality in fecal peritonitis infection. Moreover, Baghela et al. [14] found gene expression signatures that predicted prognosis with 77–80% accuracy. Interestingly, those authors suggested that patients with early sepsis could be stratified into five distinct mechanistic endotypes, based on unique gene expression differences, with variable overall severity. Some of our results concur with those of previous studies; we also found that gene expression profiles from pathways such as apoptosis and oxidative stress were differentially expressed between groups. 

In the present study, we found differences in the transcriptional profiles at hospital admission between CAP patients who died during hospitalization and those who survived. These findings are also supported by studies that have documented that the pathways that we found are related to prognoses in patients with sepsis or CAP. Some of these pathways can be regarded as “double-edged swords”: on one hand, they are useful for fighting infectious pathogens, but on the other, they are harmful and produce organ damage. Functional analysis showed that gene sets positively enriched in CAP patients who died were associated with apoptosis, interferon alpha response, and sex hormones. Regarding apoptosis, p53 is a stress-induced transcription factor that can be activated via several stimuli, including hypoxia and reactive oxygen species [15]. It has been found that inappropriate regulation of apoptosis in immune, endothelial, and pulmonary epithelial cells may play a critical role in production of immune dysfunction, impaired perfusion, tissue hypoxia, and multiple organ failure in sepsis [16]. Evidence suggests that prevention of cell apoptosis can improve prognosis in animal models of sepsis. One study found that the lungs of naïve p53(-/-) mice displayed proinflammatory genes and clear pathogens more successfully than did controls after intrapulmonary infection [17]. 

Moreover, our data also identified a significant enrichment in genes associated with spermatogenesis. Sex hormone regulation is carried out via the hypothalamic–pituitary–gonadal axis. Sex hormones have been reported to have regulatory influences on immune responses; estradiol can stimulate production of proinflammatory cytokines and macrophage activation, and testosterone has a suppressive effect on immune responses and increases vulnerability to infection [18]. High levels of estrogens such as estradiol have been observed in male and female patients with sepsis and septic shock and have been associated with significantly higher risk of in-hospital mortality [18,19]. Moreover, in males with CAP, sex and mineralocorticoid hormone metabolites have been associated with inflammation, disease severity, and long-term survival [20].

Another key feature of the gene expression profiles in CAP nonsurvivors was upregulation of the interferon-alpha response pathway. Type 1 interferon-alpha is mainly an antiviral cytokine. However, it has proven useful for control of bacterial replication and lung inflammation and improved clinical outcomes in animal models of bacterial pneumonia via increased neutrophil and macrophage activation with release of reactive oxygen and nitrogen species and bacterial killing [21]. Nevertheless, interferon-alpha can also cause pathogenic damage and an uncontrolled inflammatory response [22]. Finally, interferon regulatory factor 5 (IRF5) and its related inflammatory cytokines, such as interferon-alpha, have been associated with severity and prognoses in CAP patients [23]. 

Functional analysis showed that gene sets related to the oxidative stress, angiogenesis, and endoplasmic reticulum stress pathways were positively enriched in patients who survived. Regarding oxidative stress, organisms that live under aerobic conditions are exposed to several oxidizing agents, including reactive oxygen species (ROSs) and reactive nitrogen species (RNSs). These species perform biological functions that are essential for normal cell development; however, an imbalance between reactive-species generation and antioxidant defense, known as oxidative stress, can result in impaired homeostasis and lead to various pathologies [24]. Oxidative stress is part of the pathogenic mechanism of CAP and is closely linked to inflammation [25]. 

Numerous biomarkers have been associated with angiogenesis, including angiopoietins, members of the vascular endothelial growth factor family, transforming growth factors, interleukins, platelet-derived growth factor, and the fibroblast growth factor family. During infection, factors related to angiogenesis and the endothelial barrier are essential for migration of immune-system cells into infected tissues but can also participate in the pathogenesis of septic shock and acute multiple organ dysfunction [26]. Moreover, under conditions that cause stress and inflammation, the endoplasmic reticulum loses homeostasis in a process termed endoplasmic reticulum stress. During endoplasmic reticulum stress, an unfolded protein response (UPR) is activated to restore the normal endoplasmic reticulum function. This UPR preserves a homeostatic environment and regulates a wide variety of cell processes, such as cell proliferation and differentiation, inflammation, apoptosis, and angiogenesis. However, the UPR becomes a threat when its activation is intense and prolonged, and may lead to cell dysfunction, death, and disease [27]. Finally, transcription factor MYC may be an important regulatory gene in the underlying dysfunction of sepsis-induced acute respiratory distress syndrome (ARDS) [28].

The present pilot study has several limitations that should be acknowledged. First, the number of nonsurvivors was small, and we were unable to complete subgroup analyses; therefore, our findings need to be validated in larger cohorts from different geographical areas. Second, we did not adjust the results for confounding variables such as age or underlying diseases. Third, we measured gene expression profiles at only one point in the disease and did not evaluate changes over the course of admission; therefore, we cannot rule out the possibility that gene expression may differ at other times during CAP. In this regard, it should be noted that the findings of our study show gene expression profiles specifically in the initial phases of CAP. Finally, we did not confirm the results with real-time quantitative polymerase chain reaction of the target genes.

## 5. Conclusions

The gene expression profiles of CAP survivors and nonsurvivors presented differences, mainly related to interferon-alpha response, apoptosis, sex hormones, oxidative stress, unfolded protein response, and angiogenesis pathways. These findings may expand our understanding of the immune response in CAP through identification of new candidate pathways and targets for potential intervention. In addition, the differentially expressed genes could potentially be useful as risk-stratification biomarkers that may facilitate healthcare utilization.

## Figures and Tables

**Figure 1 biomedicines-11-00429-f001:**
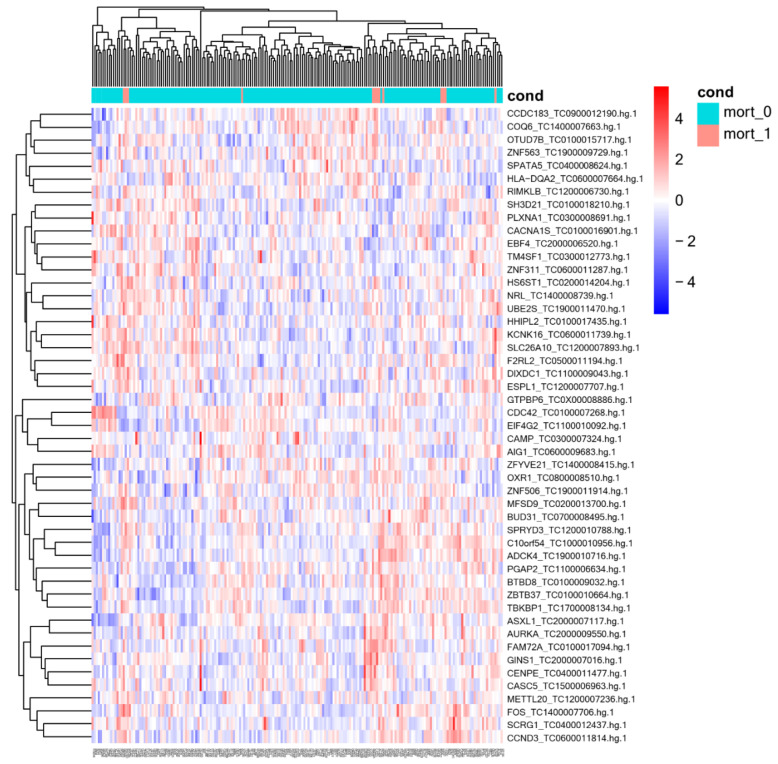
Heatmap of differentially expressed genes between community-acquired pneumonia patients who survived and died. mort_0, patients who survived; mort_1, patients who died.

**Figure 2 biomedicines-11-00429-f002:**
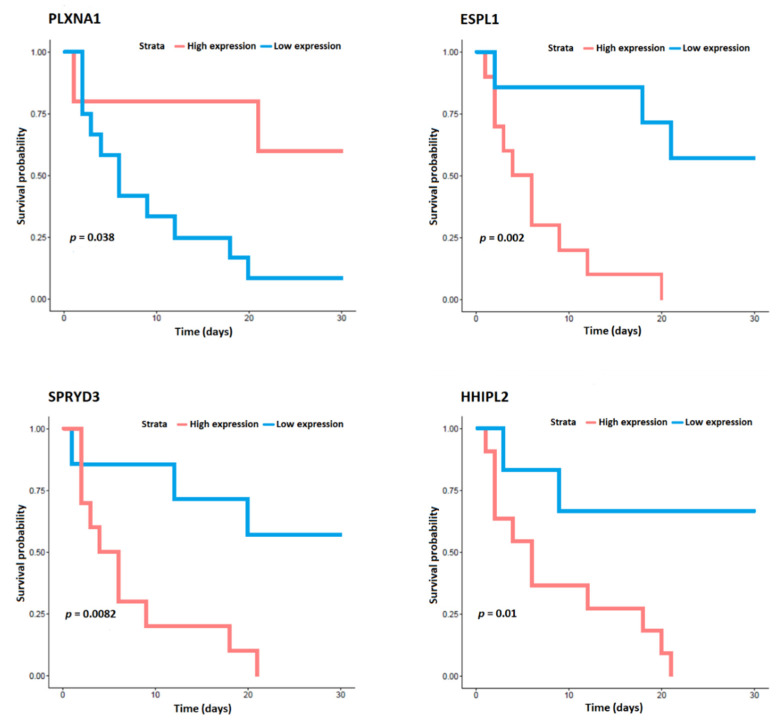
Analyses of the expression data for the genes *SPRYD3*, *ESPL1*, *HHIPL2*, and *PLXNA1* and for survival in community-acquired pneumonia patients.

**Figure 3 biomedicines-11-00429-f003:**
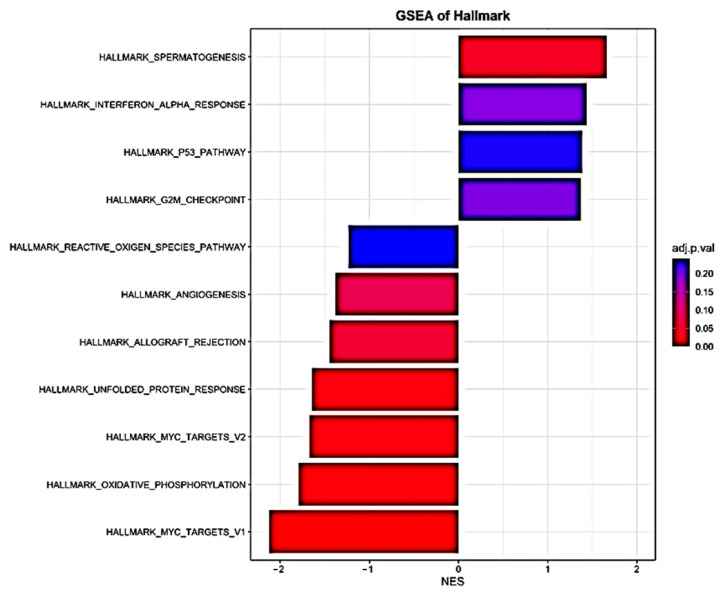
Histogram of Hallmark-gene-set enrichment analysis of transcriptional differences in whole blood associated with mortality in community-acquired pneumonia.

**Figure 4 biomedicines-11-00429-f004:**
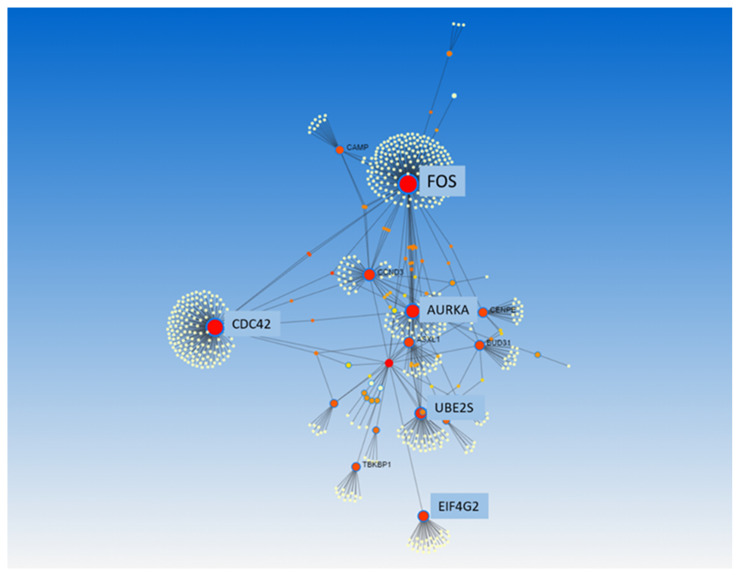
Protein–protein interaction analysis, from NetworkAnalyst 3.0, of transcriptional differences in whole blood associated with mortality in community-acquired pneumonia. All differentially expressed genes were used to perform this analysis.

**Table 1 biomedicines-11-00429-t001:** Characteristics of hospitalized patients with community-acquired pneumonia.

Characteristics	All Patients (N = 228)	Patients Who Died (N = 18)	Patients Who Survived (N = 210)	*p*-Value
**Sociodemographic Data**				
Age (years), Median (IQR)	75 (62.5–84)	84.5 (59–92)	74.5 (63–83)	0.04
Male Sex	136 (59.6)	12 (66.7)	124 (59)	0.52
Current Smoker	43 (18.9)	2 (11.1)	41 (19.5)	0.53
COPD	72 (31.6)	4 (22.2)	68 (32.4)	0.37
Chronic Heart Disease	82 (36)	9 (50)	73 (34.8)	0.19
Diabetes Mellitus	49 (21.5)	6 (33.3)	43 (20.5)	0.23
**Clinical Features at Admission**				
Time from Symptom Onset (Days), Median (IQR)	4 (2–7)	3.5 (1–5)	4 (2–7)	0.19
Fever (>38.0 °C)	84 (37)	3 (16.7)	81 (38.0)	0.06
Tachycardia (≥100 Beats × min^−1^)	135 (59.2)	11 (61.1)	124 (59)	0.86
Tachypnea (≥30 Breaths × min^−1^)	66 (34.4)	9 (60)	57 (32.2)	0.03
Impaired Consciousness	32 (14)	3 (16.7)	29 (13.8)	0.72
Septic Shock	17 (7.5)	5 (27.8)	12 (5.7)	0.001
**Laboratory and Radiographic Findings**				
Respiratory Failure (PaO_2_/FiO_2_ < 300 or PaO_2_ < 60 mmHg)	130 (57)	15 (83.3)	115 (54.8)	0.019
Leukocytosis (Leukocytes ≥ 12 × 10^9^/L)	136 (59.6)	12 (66.7)	124 (59)	0.52
Multilobar Pneumonia	71 (31.1)	6 (33.3)	65 (32.3)	0.93
Pleural Effusion	8 (3.5)	2 (11.1)	26 (12.4)	1
Bacteremia	19 (8.9)	3 (16.7)	16 (8.2)	0.20
Bacterial Pneumonia	82 (36)	6 (33.3)	76 (36.2)	0.80
Pneumococcal Pneumonia	48 (21.1)	3 (16.7)	45 (21.4)	0.77
Viral Pneumonia	12 (5.3)	1 (5.6)	11 (5.2)	1
**CAP-Specific Scores**				
PSI Score, Median (IQR)	100 (81.5–123.5)	142 (109–178)	99 (79–121)	<0.001
PSI High-Risk Classes (IV–V)	146 (64)	17 (94.4)	129 (61.4)	0.005

COPD, chronic obstructive pulmonary disease; ICU, intensive care unit; IQR, interquartile range; PSI, pneumonia severity index.

**Table 2 biomedicines-11-00429-t002:** List of genes differentially expressed in community-acquired pneumonia patients who died in relation to patients who survived based on the lowest *p*-value.

Symbol	Gene Name	*p*-Value	Fold Change	Resampling (N Times)
*SCRG1*	Stimulator of chondrogenesis 1	0.001	1.28	383
*FAM72A*	Family with sequence similarity 72, member A	0.001	1.45	363
*KCNK16*	Potassium channel, two-pore-domain subfamily K, member 16	0.001	1.20	270
*SLC26A10*	Solute carrier family 26, member 10	0.001	1.25	267
*ZNF563*	Zinc finger protein 563	0.001	1.18	448
*OTUD7B*	OTU deubiquitinase 7B	0.002	1.20	339
*ADCK4*	aarF domain containing kinase 4	0.004	1.16	533
*F2RL2*	Coagulation factor II (thrombin) receptor-like 2	0.004	1.18	337
*GTPBP6*	GTP binding protein 6 (putative)	0.004	−1.13	286
*TBKBP1*	TBK1 binding protein 1	0.004	1.35	365
*HHIPL2*	HHIP-like 2	0.005	1.14	346
*PLXNA1*	Plexin A1	0.005	−1.13	339
*SPATA5*	Transcript Identified with AceView, Entrez Gene ID(s) 166378	0.006	−1.18	418
*NRL*	Neural retina leucine zipper	0.006	1.17	317
*UBE2S*	Ubiquitin-conjugating enzyme E2S	0.006	1.18	268
*MFSD9*	Major facilitator superfamily domain containing 9	0.007	1.28	327
*OXR1*	Oxidation resistance 1	0.007	1.24	257
*DIXDC1*	DIX domain containing 1	0.007	1.23	344
*SH3D21*	SH3 domain containing 21	0.009	−1.16	337
*PGAP2*	Post-GPI attachment to proteins 2	0.009	1.18	265
*FOS*	FBJ murine osteosarcoma viral oncogene homolog	0.009	1.19	318
*ZFYVE21*	Zinc finger, FYVE domain containing 21	0.009	1.18	261
*HS6ST1*	Heparan sulfate 6-O-sulfotransferase 1	0.009	1.12	258
*BTBD8*	BTB (POZ) domain containing 8	0.012	1.22	315
*RIMKLB*	Ribosomal modification protein rimK-like family member B	0.012	1.19	312
*EBF4*	Early B-cell factor 4	0.012	−1.14	428
*METTL20*	Methyltransferase-like 20	0.013	1.16	293
*EIF4G2*	Eukaryotic translation initiation factor 4 gamma, 2	0.016	−1.15	279
*CACNA1S*	Calcium channel, voltage-dependent, L type, alpha 1S subunit	0.018	−1.14	304
*TM4SF1*	Transmembrane 4 L six family member 1	0.023	−1.19	357
*ZNF311*	Zinc finger protein 311	0.023	−1.12	268
*HLA-DQA2*	Major histocompatibility complex, class II, DQ alpha 2	0.026	−1.24	255
*ESPL1*	Extra spindle pole bodies like 1, separase	0.028	1.12	266
*BUD31*	Transcript Identified with AceView, Entrez Gene ID(s) 8896	0.030	1.17	283
*CCND3*	Memczak2013 ALT_ACCEPTOR, ALT_DONOR, coding, INTERNAL, intronic best transcript NM_001136017	0.031	1.34	318
*GINS1*	GINS complex subunit 1 (Psf1 homolog)	0.031	1.19	262
*ASXL1*	Additional sex combs like transcriptional regulator 1	0.031	1.21	404
*C10orf54*	Chromosome 10 open reading frame 54	0.033	1.18	403
*AURKA*	Aurora kinase A	0.034	1.18	332
*ZNF506*	Zinc finger protein 506	0.035	1.13	267
*CASC5*	Cancer susceptibility candidate 5	0.035	1.18	261
*AIG1*	Androgen-induced 1	0.039	−1.17	303
*SPRYD3*	SPRY domain containing 3	0.039	1.13	391
*CDC42*	Cell division cycle 42	0.039	−1.09	323
*ZBTB37*	Zinc finger and BTB domain containing 37	0.041	1.19	258
*CAMP*	Cathelicidin antimicrobial peptide	0.043	−1.46	304
*CCDC183*	Coiled-coil domain containing 183	0.044	1.10	343
*COQ6*	Coenzyme Q6 monooxygenase	0.046	1.13	252
*CENPE*	Centromere protein E	0.049	1.16	312

**Table 3 biomedicines-11-00429-t003:** Gene set enrichment analysis of transcriptional differences in whole blood associated with mortality in community-acquired pneumonia.

Gene Set Name	NOM *p*-Value	FDR q-Value	NES	Brief Description
**Positive Enrichment Score**				
HALLMARK_SPERMATOGENESIS	0.000	0.034	1.66	Genes upregulated during production of male gametes (sperm), as in spermatogenesis.
HALLMARK_INTERFERON_ALPHA_RESPONSE	0.011	0.202	1.44	Genes upregulated in response to alpha interferon proteins.
HALLMARK_P53_PATHWAY	0.016	0.237	1.39	Genes involved in p53 pathways and networks.
HALLMARK_G2M_CHECKPOINT	0.020	0.199	1.37	Genes involved in the G2/M checkpoint, as in progression through the cell division cycle.
**Negative Enrichment Score**				
HALLMARK_MYC_TARGETS_V1	0.000	0.000	−2.13	A subgroup of genes regulated with MYC—version 1 (v1).
HALLMARK_OXIDATIVE_PHOSPHORYLATION	0.000	0.005	−1.80	Genes encoding proteins involved in oxidative phosphorylation.
HALLMARK_MYC_TARGETS_V2	0.000	0.010	−1.67	A subgroup of genes regulated with MYC—version 2 (v2).
HALLMARK_UNFOLDED_PROTEIN_RESPONSE	0.000	0.010	−1.65	Genes upregulated during unfolded protein response, a cellular stress response related to the endoplasmic reticulum.
HALLMARK_ALLOGRAFT_REJECTION	0.000	0.051	−1.45	Genes upregulated during transplant rejection.
HALLMARK_ANGIOGENESIS	0.072	0.076	−1.39	Genes upregulated during formation of blood vessels (angiogenesis).
HALLMARK_REACTIVE_OXYGEN_SPECIES_PATHWAY	0.139	0.241	−1.24	Genes upregulated by reactive oxygen species (ROS).

FDR, false discovery rate; NOM, nominal *p*-value; NES, normalized enrichment score.

**Table 4 biomedicines-11-00429-t004:** Functional roles for hub genes of transcriptional differences in whole blood associated with mortality in community-acquired pneumonia.

Symbol	Gene Name	Function
*FOS*	FBJ murine osteosarcoma viral oncogene homolog	Cell proliferation, differentiation, and transformation. In some cases, also associated with apoptotic cell death. Among its related pathways are the IL-6 and Toll-like receptor signaling pathways.
*CDC42*	Cell division cycle 42	Controls diverse cellular functions, including cell morphology, migration, endocytosis, and cell-cycle progression. Also plays a role in phagocytosis, thymocyte development, T cell actin and tubulin cytoskeleton polarization, and T cell migration.
*AURKA*	Aurora kinase A	Mitotic serine/threonine kinase that contributes to regulation of cell-cycle progression.
*UBE2S*	Ubiquitin-conjugating enzyme E2S	An essential factor of the anaphase-promoting complex/cyclosome (APC/C), a cell-cycle-regulated ubiquitin ligase that controls progression through mitosis.
*CCND3*	Memczak2013 ALT_ACCEPTOR, ALT_DONOR, coding, INTERNAL, intronic best transcript NM_001136017	Regulatory component of the cyclin D3-CDK4 (DC) complex that phosphorylates and inhibits members of the retinoblastoma (RB) protein family, including RB1, and regulates the cell cycle during G(1)/S transition.
*SLC26A10*	Solute carrier family 26, member 10	Diseases associated with SLC26A10 include sialolithiasis and Pendred syndrome. Antiporter activity and sulfate transmembrane transporter activity.
*EIF4G2*	Eukaryotic translation initiation factor 4 gamma, 2	Appears to play a role in the switch from cap-dependent to IRES-mediated translation during mitosis, apoptosis, and viral infection. Cleaved with some caspases and viral proteases.
*ASXL1*	Additional sex combs-like transcriptional regulator 1	Determination of segment identity in the developing embryo. Necessary for the maintenance of stable repression of homeotic and other loci. Enhances transcription of certain genes while repressing transcription of others.

## Data Availability

The original contributions presented in this study are included in the article/Appendix A. The microarray data from the project have been deposited in NCBI’s Gene Expression Omnibus under accession number GSE188309. Further inquiries can be directed to the corresponding authors.

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
