# Peer review of "Whole-Blood Gene Expression Profiles Associated with Mortality in Community-Acquired Pneumonia"

_biomedicines, 2023, doi:10.3390/biomedicines11020429_

Round 1
Reviewer 1 Report
In the present study, Viasus et al. examined the differences at hospital admission in gene expression profiles in peripheral blood between community acquired pneumonia in patients who subsequently died and those who survived. Their scope was to identify genes and related pathways that not only provide information about pathophysiology but may also serve as prognostic biomarkers.
Their work is interesting and has merit for publication, yet after addressing some major issues.
"Introduction"
please provide some background on the mechanisms and bacteria causing pneumonia. Not all readers are familiar with the details of the mechanism.
"Materials and Methods"
Please provide statistical methods for data analysis (see comments on results below).
Please describe the linear models for identifying differentially expressed genes (DEGs).
"Results"
This is the major issue of the present work. The authors should present some additional statistical analysis concerning their data.
First of all, the authors should present a hierarchical clustering (HCL) of their data, i.e. DEGs.
Second, they should present a statistical analysis (for example ANOVA, which should be also added to the "Materials and Methods" section) of gene expression with respect to the patients' phenotype (e.g. disease outcome, comorbidities etc.).
Third, they should attempt to present a ROC curve in order to show if the identified genes were any good in describing disease outcome, prognosis, course etc.
"Discussion"
the authors should highlight their findings and mention the use of their findings in the disease and how they can applied for disease prognosis, therapy.
Reviewer 2 Report
Title: Whole blood gene expression profiles associated with mortality in community-acquired pneumonia
Summary: In this study authors performed gene expression analysis on whole blood for CAP patients. Authors then determined association of enriched genes with patient mortality. The study looks good overall. However, authors should address the following concerns:
Major comments:
1. Supplementary Fig 1. Heatmap is an important figure. Authors should move it to Main Figure 1.
2. Table 2: Authors should divide the presentation of DEGs into upregulated and downregulated genes and also provide expression value for each gene in the table. Only p value is not enough information to understand which genes are up or down regulated.
3. Authors should move Survival plots for genes SPRYD3, ESPLI ,HHIPL2, and PLXNA1 to main figure.
4. Fig 2: It is unclear, which set of genes were used to perform PPI analysis. Authors should clarify in results section whether they used up- or down regulated genes or all DEGs to perform this analysis.
Round 2
Reviewer 2 Report
Major comments:
1. Authors state in the author response letter that they have added Fig 1 heatmap to the main figure, but they have not added heatmap to the main figure. Only legend is there. Please add the missing figure to manuscript.
2. Authors state in the author response letter that they have added Fig 2 survival plot for genes SPRYD3, ESPLI ,HHIPL2, and PLXNA1 to main figure, but they have not added any survival plots to the main figure. Only legend is there. Please add the missing figure to manuscript.
Author Response
Barranquilla, January 24, 2023
Reviewer 2
Major comments:
- Authors state in the author response letter that they have added Fig 1 heatmap to the main figure, but they have not added heatmap to the main figure. Only legend is there. Please add the missing figure to manuscript.
We have added heatmap to the manuscript (Figure 1).
- Authors state in the author response letter that they have added Fig 2 survival plot for genes SPRYD3, ESPLI ,HHIPL2, and PLXNA1 to main figure, but they have not added any survival plots to the main figure. Only legend is there. Please add the missing figure to manuscript.
We have added this figure to the manuscript (Figure 2).

Round 3
Reviewer 2 Report
Authors have responded to the comments. Manuscript is acceptable as it stands.